# Understanding the Pathogenesis of Red Mark Syndrome in Rainbow Trout (*Oncorhynchus mykiss*) through an Integrated Morphological and Molecular Approach

**DOI:** 10.3390/ani13061103

**Published:** 2023-03-20

**Authors:** Marco Galeotti, Massimo Orioles, Elena Saccà, Omkar Byadgi, Stefano Pesaro, Alessandro Di Cerbo, Gian Enrico Magi

**Affiliations:** 1Department of Agricultural, Food, Environmental and Animal Sciences, DI4A, University of Udine, 33100 Udine, Italy; 2International Program in Ornamental Fish Technology and Aquatic Animal Health, International College, National Pingtung University of Science and Technology, No. 1, Shuefu Road, Neipu, Pingtung 91201, Taiwan; 3School of Biosciences and Veterinary Medicine, University of Camerino, 62024 Matelica, Italy

**Keywords:** *Oncorhynchus mykiss*, Red Mark syndrome, *Midichloria*-like organism, gene expression, Quantitative PCR, host immune response

## Abstract

**Simple Summary:**

Red mark syndrome is a non-lethal widespread skin disease mainly reported in rainbow trout and caused by a *Midichloria*-like organism. Despite extensive research, its etiology and pathogenesis are still uncertain. In the present study, the authors used an integrated morphological and molecular approach, including gene expression, to elucidate the immune response and the complex immune interaction between the host and *Midichloria*-like organism. The results lead to the conclusions that the most severe skin lesions were characterized by a high level of inflammatory cytokines sustaining and modulating the severe inflammatory process. In contrast, in the moderate form, the response was driven to produce immunoglobulins and IL-10 to control the severity of the disease. Humoral immunity elicited during MLO infection appeared to have a fundamental role in controlling the severity of the skin disease, possibly through bactericidal antibody-mediated mechanisms.

**Abstract:**

Red mark syndrome (RMS) is a widespread skin disorder of rainbow trout in freshwater aquaculture, believed to be caused by a *Midichloria*-like organism (MLO). Here, we aimed to study the pathologic mechanisms at the origin of RMS by analyzing field samples from a recent outbreak through gene expression, MLO PCR, quantitative PCR, and a histopathological scoring system proposed for RMS lesions. Statistical analyses included a One-Way Analysis of Variance (ANOVA) with a Dunnett’s multiple comparisons test to assess differences among gene expression groups and a nonparametric Spearman correlation between various categories of skin lesions and PCR results. In short, the results confirmed the presence of a high quantity of 16S gene copy numbers of *Midichloria*-like organisms in diseased skin tissues. However, the number of *Midichloria*-like organisms detected was not correlated to the degree of severity of skin disease. *Midichloria*-like organism DNA was found in the spleen and head kidney. The spleen showed pathologic changes mainly of hyperplastic type, reflecting its direct involvement during infection. The most severe skin lesions were characterized by a high level of inflammatory cytokines sustaining and modulating the severe inflammatory process. IL-1 β, IL-6, IL-10, MHC-II, and TCR were upregulated in severe skin lesions, while IL-10 was highly expressed in moderate to severe ones. In the moderate form, the response was driven to produce immunoglobulins, which appeared crucial in controlling the skin disease’s severity. Altogether our results illustrated a complex immune interaction between the host and *Midichloria*-like organism.

## 1. Introduction

Red mark syndrome (RMS) is a non-lethal skin disorder affecting rainbow trout (*Oncorhynchus mykiss*) in freshwater aquaculture. It causes bright red skin lesions leading to marked economic losses on behalf of the fish farmer with high morbidity (up to 90%) without mortality, with no effects on weight, appetite, or fish behavior [1]. Its development and spontaneous resolution seem markedly dependent upon water temperature [2,3]. RMS condition in rainbow trout has been widely reported in at least three continents [4,5,6,7,8,9,10,11,12,13,14,15,16,17,18]. This disease seems transmissible, and current literature suggests a bacterial infection as the likely etiology [2,3,19,20,21,22], with two agents, Flavobacterium psychrophilum and a Rickettsia-like organism, considered the most likely candidates. Further investigation demonstrated a robust and favorable correlation between MLO and RMS-affected fish [22]. No meaningful association was observed between RMS-affected fish and F. psychrophilum, except for its superficial presence on the skin [22]. Thus, to date, a *Rickettsia*-like organism (RLO), belonging to the Midichloriaceae family and the order of Rickettsiales, and named *Midichloria-like* organism (MLO) is believed to be the cause of RMS [1,23,24]. This has been supported by many studies where MLOs were visualized both in splenic impression smears as morula-like structures, typical of *Rickettsia*-like organisms, in the cytoplasm of splenic macrophages of RMS-affected rainbow trout [18,25], and by TEM as intracytoplasmic microorganisms resembling Rickettsiales within macrophages, fibroblasts, and erythrocytes both in the field and experimental cases [3,24]. Furthermore, the application of immunohistochemistry showed positive staining of what was considered MLO-related antigens in the internal organs of most RMS-affected fish [22].

To date, the diagnosis of RMS is typically based on clinical features and histopathology, complemented by molecular biology techniques, such as PCR for MLO DNA [1,26]. A recent study proposed by the authors provides a histological scoring system based on three categories of severity of microscopic lesions [26]. The loss of epithelium and scales, the presence of inflammatory cells in the stratum compactum, disruption of the architecture of the stratum compactum, granulomatous inflammation around blood vessels and nerves, and an increase in lymphocytes in the muscular layer were all significant histological parameters that correlated with the severity of gross lesions.

The pathogenesis of RMS is a subject of ongoing debate among the scientific community. Previous studies that have sought to elucidate this issue have primarily focused on skin tissue [21,27]. Nonetheless, these studies have several limitations, such as failing to examine other organs, including the spleen and kidney, and needing more comprehensive information on MLO PCR [27] and a clearly defined histological scoring system for RMS lesions [21,27]. Despite these limitations, these investigations have suggested that the fish host may mount a Th1-type response to overcome the infection, aided by the possible T cell-independent production of antibodies [21]. Consequently, RMS has been classified as an immunopathological syndrome [27]. In addition to those conclusions, the appearance of microthrombosis and necrosis of the epidermis in advanced severe lesions was comparable to that found in skin inflammation reactions caused by the buildup of immune complexes (type III). Furthermore, these immune complexes can coincide with delayed-type hypersensitivity (type IV), as seen in canine scabies, where both type III and type IV mechanisms are present [26].

To advance our understanding of the pathogenic mechanisms underlying RMS, our study aims to investigate the host immune response by utilizing a histopathological scoring system for skin lesions [26] while examining other organs, such as spleen and kidney tissues, which may be implicated in the disease’s pathogenesis. To detect the MLO copy number, gene expression analysis and a biomolecular approach, including MLO-specific and quantitative PCR, were used.

## 2. Materials and Methods

### 2.1. Outbreaks and Fish Tissue Sampling

The outbreak under study occurred in two intensive rainbow trout farms during 2020, late winter and spring. Fish were reared to commercial size in concrete basins supplied by river water (water temperature 8–11 °C) and fed with a commercial pelleted diet. According to the case definition and recent literature, rainbow trout showed typical signs of disease compatible with RMS [26,28]. The percentage of affected fish in total was between 20 and 30%, and their size was 250–300 g. There was no mortality or impact on the growth or feed conversion rate. Thirty-five fish with typical signs were euthanized by an overdose of MS222 and sent for assessment at the University of Udine the same day. To be used as reference samples, five healthy fish were collected from a commercial rainbow trout farm, which has never reported cases of RMS and was considered an RMS-free facility. The study was conducted according to the guidelines of and approved by the Ethics Committee of the University of Udine (Approval Code: Protocol Number 9/2022).

Skin, spleen, and head kidney were sampled from each fish. Samples were immersed in ethanol and preserved at −20 °C for DNA extraction and in RNAlater solution (Ambion, Thermo Fisher Scientific, Waltham, MA, USA) for total RNA extraction. Tissue samples for RNA analysis were kept at 4 °C for 24 h and then stored at −80 °C until further use. For the histological examination, tissue samples obtained from the skin, spleen, and kidney were fixed in 4% neutral buffered formaldehyde and then processed by an automatic histoprocessor (TISBE, Diapath, Bergamo, Italy) to be embedded in paraffin (ParaplastPlus, Diapath, Bergamo, Italy). Serial 5 μm sections were stained with hematoxylin–eosin (HE).

### 2.2. Morphological Analysis

Thirty-five fish sampled from both farms were morphologically scored for macroscopic and microscopic skin lesions. Skin lesions were grossly included in one of three categories according to the progressive degree of severity as described by recent literature [26]: type I (“Mild”), type II (“Moderate”), and type III (“Severe”). Type I included lesions up to 1 cm in size, with pale grey to whitish color, with mild exudate and visible scale; type II included lesions of 1 to 2 cm in size, from grey to reddish in color, with petechiae, moderate exudate, and mild scale loss; type III included lesion of more than 2 cm in size, with marked redness and hemorrhages, moderate to severe exudate, evident scale loss and frequent erosion of epidermis. The skin lesion distribution was also classified as focal if a single lesion was present or multifocal if multiple lesions were observed. Where more than one lesion was observed, the most severe was evaluated for classification. Histologically, skin lesions were classified into three progressive categories of severity (I “Mild”, II “Moderate” and III “Severe”) considering: the degree and extent of inflammation, vascular involvement from congestion to hemorrhages, and thrombosis of vessel, erosion, and ulceration of epidermis [26]. Spleen and kidney tissues were grossly and histologically examined for pathological changes. Histologically, splenic tissue was classified into three categories depending on the severity of ellipsoid hyperplasia, hemosiderosis, and vascular changes. Category I (called “Normal”) identified normal splenic tissue, category II (defined “Hyperplastic”) was characterized by hyperplasia of ellipsoids, and category III (named “Hyperplastic with congestion/haemorrhage and hemosiderosis”) presented hyperplastic changes associated with vascular changes and hemosiderosis.

### 2.3. DNA Extraction

All skin, spleen, and head kidney samples undergoing DNA extraction were stored in ethanol and processed using the QIAamp DNA Mini kit (Qiagen, Hilden, Germany) according to the manufacturer’s instructions for animal tissue. Samples were lysed (3 h) before washing and extraction were completed. The integrity of the extracted DNA was evaluated in 2% agarose gel, which verified the presence of a single band of intact DNA. Total genomic DNA was assessed for yield and purity using a NanoDrop One spectrophotometer (Thermo Fisher Scientific, Waltham, MA, USA). DNA samples were stored at −20 °C until further use.

### 2.4. Polymerase Chain Reaction (PCR) of MLO DNA

The specific MLO DNA was amplified using a nested PCR, according to Galeotti et al., 2017b [24], with few modifications. Every PCR amplification was carried out in a final volume of 50 µL. First-step PCR was carried out using RLO1/RLO2 [19] primers in a Bio-Rad thermocycler (Bio-Rad CA, USA) with a reaction mixture containing 100 ng DNA, 100 ng of each primer, and 1.25 Units of Taq DNA polymerase (Thermo Fisher Scientific, Waltham, MA, USA).under the following conditions: denaturation at 95 °C for 5 min, followed by 35 cycles of 95 °C for 30 s, 69 °C for 30 s (annealing), 72 °C for 30 s, and a final extension at 72 °C for 10 min. For each PCR, water has been used as a blank (negative control) to verify the absence of contaminations. A second-step PCR assay was performed using the primers pair RiFCfw 5′-AAGGCAACGATCTTTAGTTGG-3′ and RiFCrev 5′-CCGTCATTATCTTCCCCACT-3′ [24]. The amplification was conducted using 2 µL of the first step of PCR as a template, following the protocol: 95 °C denaturation for 5 min, 35 cycles of 95 °C for 45 s, 54 °C for 45 s, 72 °C for 45 s, and a final extension at 72 °C for 7 min. Subsequently, 5 µL of the PCR product were analyzed by 2% agarose gel electrophoresis stained with ethidium bromide and visualized with a UV transilluminator. PCR product size molecular weight was estimated using a 50 bp DNA ladder (Thermo Fisher Scientific, Waltham, Massachusetts, USA). To identify the type of MLO isolate in the present study, PCR products were purified using a QIAquick Purification kit (Qiagen, Hilden, Germany) and directly sequenced using the RiFCfw/RiFCrev primer set by Eurofins Genomics, Ebersberg, Germany. The sequences obtained using a BLAST search were compared with those published in GenBank.

### 2.5. MLO DNA Absolute Quantification

Real-time PCR assays were used to quantify the abundance of MLOs in the sampled tissues. MLO 16S rDNA, and IGF as a calibrator, were quantified from extracted DNA of skin, spleen, and head kidney samples following a previously published protocol [23], but with some modifications. Initially, a fragment of 16s rDNA of MLO and IGF was amplified using PCR; the PCR product was cloned using pGEM-T easy vector system (Promega, Madison, WI, USA) and quantified using a NanoDrop One spectrophotometer (Thermo Fisher Scientific, Waltham, Massachusetts, USA). The plasmid was purified using QIAprep Spin Miniprep Kit (Qiagen, Hilden, Germany) and quantified. The cloned plasmid was sequenced using M13 primer to confirm the insert and alignment (Eurofins Genomics, Ebersberg, Germany). Subsequently, the plasmid was linearized downstream by restriction digestion using PstI (Thermo Fisher Scientific, Waltham, MA, USA). To calculate the 16S rDNA copies number corresponding to the cycle threshold (Ct) values, the base pair length of the amplicon fragment and vector, along with DNA concentration, were submitted into a DNA-copy number calculator.

A 10-fold serial dilution of plasmid constructed was carried out in triplicate, and each log dilution was subjected to real-time amplification to detect the Ct. Only dilutions that provided Ct values in all replicates were used to generate a standard curve created by plotting the Ct values against the log-transformed copy number of each 10-fold dilution of MLO and IGF. The amplification efficiency of the real-time PCR assay was established based on the Ct slope method (Efficiency (E) = (10^(−1/slope)^) − 1), and the linearity was determined as the coefficient of correlation (R^2^). Similarly, the SYBR-based (Bio-Rad, Milano, Italy) real-time PCR was performed on DNA samples in triplicates using tissues from the two infected farms. The cycling conditions for PCR were as follows: 10 min at 95 °C followed by 40 cycles of 30 s at 95 °C and 60 s at 60 °C, after which a melt curve analysis was performed (1 min at 95 °C, 30 s at 55 °C, and incremental temperature increase to 95 °C).

### 2.6. Gene Expression Analysis

RNA extraction was performed using RNeasy Fibrous Tissue Mini Kit (Qiagen, Hilden, Germany), according to the manufacturer’s protocol, deoxyribonuclease (DNase) treatment included. Disruption and homogenization of tissue samples were performed using mortar, pestle, and liquid Nitrogen for the skin and TissueLyser II (Qiagen, Hilden, Germany) for the spleen and head kidney. The NanoDrop One spectrophotometer (Thermo Scientific, Waltham, MA, USA) was utilized to evaluate the degree of concentration and purity of the extracted RNA. RNA integrity and the absence of genomic DNA were evaluated by running 1% agarose gel electrophoresis stained with ethidium bromide. MW standards were used for comparison to highlight the presence of distinct and clear 18S and 28S rRNA bands. The iScript cDNA Synthesis kit (Bio-Rad, Milano, Italy) was used to generate complementary DNA (cDNA), following the manufacturer’s guidelines. A 20 μL reaction mix was prepared for each sample, containing 4 μL of reverse transcription reaction mix, 1 μL of iScript reverse transcriptase, and the necessary volume of RNA solution to achieve a final concentration of 50 ng/μL. Nuclease-free water made up the remaining volume. The mixture was then incubated at 25 °C for 5 min, 46 °C for 20 min, and 95 °C for 1 min, before being cooled on ice. Qualitative polymerase chain reaction (PCR) was carried out on a sample pool to validate primer pair specificity for all target genes (cytokines: Interleukin 1 beta (IL-1β), Interleukin 6 (IL-6), Interleukin 8 (IL-8), Interleukin 10 (IL-10), Tumor Necrosis Factor alpha (TNF-α); cell receptors: Toll-Like Receptor 5 (TLR5), T-Cell Receptor beta chain (TCR-β), Major Histocompatibility Complex I (MHC-I) b antigen, Major Histocompatibility Complex II (MHC-II) beta chain; immunoglobulins: Immunoglobulin M (IgM) heavy chain, Immunoglobulin T (IgT) heavy chain; reference genes: Beta-actin (B-act), Elongation Factor 1 alpha (EF1), 60S ribosomal protein (60S) (Table 1). PCR amplification was performed using the Bio-Rad CFX96 system (Bio-Rad, Milano, Italy), with a reaction volume of 20 μL. The reaction mixture consisted of 0.6 μL each of forward and reverse primers (0.3 μmol/L), 10 μL of SsoAdvanced Universal SYBR Green Supermix (Bio-Rad, Milano, Italy), 7.8 μL of sterile water, and 1 μL of cDNA. The amplification protocol comprised an initial cycle of 30 s at 95 °C, followed by 40 cycles of 15 s at 95 °C and 30 s at 60 °C. A melt curve analysis was performed between 65–95 °C with increments of 0.5 °C every 5 s to confirm the absence of nonspecific amplification and primer-dimer formation. Amplicon length (Table 1) was verified by 2% agarose ethidium bromide-stained gel electrophoresis by comparing it with molecular weight standards.

Quantitative PCR (qPCR) was performed using the same instrument, reagents, and volumes as those for qualitative PCR, and each sample was analyzed in duplicate. The primer calculation was performed using the standard curve derived from the amplification of the serial dilution of the pooled cDNA [29] (Table 1).

**Table 1 animals-13-01103-t001:** Primer sequences, amplification products, and amplification efficiency for target and reference genes.

Gene	Primers Sequence, 5′ to 3′	Amplicon Length (bp)	Accession Number (NCBI; GenBank)	Efficiency (%)	R^2^
IL-1β ^1^	F: ACATTGCCAACCTCATCATCG R: TTGAGCAGGTCCTTGTCCTTG	91	AJ223954	101.5	0.999
IL-6 ^2^	F: ACTCCCCTCTGTCACACACC R: GGCAGACAGGTCCTCCACTA	91	DQ866150	101.8	0.984
IL-8 ^3^	F: AGAATGTCAGCCAGCCTTGT R: TCTCAGACTCATCCCCTCAGT	69	AJ279069	103.0	0.994
IL-10 ^4^	F: CGACTTTAAATCTCCCATCGAC R: GCATTGGACGATCTCTTTCTTC	70	AB118099	100.0	0.991
TNF-α ^5^	F: GGGGACAAACTGTGGACTGA R: GAAGTTCTTGCCCTGCTCTG	66	AJ277604	104.0	0.997
TLR5 ^6^	F: GGCATCAGCCTGTTGAATTT R: ATGAAGAGCGAGAGCCTCAG	89	AB091105	101.0	0.987
TCR-β ^7^	F: CTCCGCTAAGGAGTGTGAAGATAG R: CAGGCCATAGAAGGTACTCTTAGC	412	AF329700	102.2	0.995
MHC-I ^8^	F: TCCCTCCCTCAGTGTCT R: GGGTAGAAACCTGTAGCGTG	73	AY523661	102.5	0.998
MHC-II ^9^	F: TGCCATGCTGATGTGCAG R: GTCCCTCAGCCAGGTCACT	67	AF115533	100.9	0.999
IgM ^10^	F: CTTGGCTTGTTGACGATGAG R: GGCTAGTGGTGTTGAATTGG	72	S63348	97.9	1.000
IgT ^11^	F: AGCACCAGGGTGAAACCA R: GCGGTGGGTTCAGAGTCA	72	AY870265	101.2	0.998
B-act ^12^	F: ACAGACTGTACCCATCCCAAAC R: AAAAAGCGCCAAAATAACAGAA	167	AJ438158	99.8	0.999
EF1 ^13^	F: ACCCTCCTCTTGGTCGTTTC R: TGATGACACCAACAGCAACA	63	AF498320	100.0	0.999
60S ^14^	F: AGCCACCAGTATGCTAACCAGT R: TGTGATTGCACATTGACAAAAA	147	NM001165047	104.1	0.998

IL-1β: Interleukin 1 beta; IL-6: Interleukin 6; IL-8: Interleukin 8; IL-10: Interleukin 10; TNF-α: Tumour Necrosis Factor alpha; TLR5: Toll-Like Receptor 5; MHC-I: Major Histocompatibility Complex I b antigen; MHC-II: Major Histocompatibility Complex II beta chain; TCR-β: T-Cell Receptor beta chain; IgM: Immunoglobulin M heavy chain; IgT: Immunoglobulin T heavy chain; B-act: Beta-actin; EF1: Elongation Factor 1 alpha; 60S: 60S ribosomal protein. References: ^1,4,8,9,10,11,13^ [30]; ^2,6^ [31]; ^3,5,12,14^ [32]; ^7^ [33]. F: forward, R: reverse.

### 2.7. Statistical Analysis

CFX Maestro software (version 2.2, Bio-Rad, Milano, Italy) was used to evaluate the stability of the reference genes. To normalize sample data, the geometric mean of the most stable gene between B-act, EF1, and 60S was used for each elaboration [34]. Relative gene expression was calculated using the efficiency-corrected 2^−ΔΔCt^ method [29,35]. All data were presented as the means ± standard deviation (SD). A One-Way Analysis of Variance (ANOVA) with a Dunnett’s multiple comparisons test was used to analyze differences among the gene expression of the considered groups in skin, spleen, and head kidney. The gene expression data in skin related to the gravity of the microscopical lesions of the tissue were divided into three groups, “Mild”, “Moderate”, and “Severe”, and data were presented as the fold-change ratio with the “Mild” group as reference. The data of gene expression in the spleen related to the gravity of the microscopical lesions of the tissue were divided into three groups, “Normal”, “Hyperplasia” and “Hyperplasia with congestion/hemorrhage and hemosiderosis”, and data were presented as the fold-change ratio with the “Normal” group as reference. The data of gene expression in skin, related to the concentration of MLO DNA in the tissue, were divided into three groups, respectively, “Low”, “Medium” and “High”, and data were presented as the fold-change ratio with the “Low” group as reference. The gene expression data in the spleen, related to the concentration of MLO DNA in the tissue, were divided into three groups, “Absent”, “Low” and “Medium”, and data were presented as the fold-change ratio with the “Absent” group as reference. The data of gene expression in the head kidney, related to the concentration of MLO DNA in the tissue, were divided into three groups, “Very Low”, “Low” and “Medium”, and data were presented as the fold-change ratio with the “Very Low” group as reference. A Friedman test followed by a Dunn’s multiple comparison test was used to analyze differences among the absolute quantity of MLO DNA of skin, spleen, and head-kidney. Possible correlation among macroscopic and microscopic skin lesions, quantitative PCR of spleen, skin, and head-kidney, as well as focal or multifocal macroscopic lesions and PCR positivity, was assessed using a nonparametric Spearman correlation. All statistical analyses were performed with GraphPad Prism 8 (GraphPad Software Inc., San Diego, CA, USA). A *p* < 0.05 was considered significant.

## 3. Results

### 3.1. Gross Lesions and Histology of Skin, Spleen, and Head Kidney

The skin exhibited gross, singular, or multiple lesions, varying in sizes from small to large foci measuring 1 × 1 to 3 × 4 cm. These lesions were flat or protruding, displaying a round or oval shape, and appeared in pink or pale red shades. Occasionally, there were scattered petechial hemorrhages, and some had desquamation in the central part of the lesion. Furthermore, the lesions were frequently covered by serous or fibrin exudate (Figure 1a–c). According to the criteria adopted for grading the macroscopic skin lesion, 8 cases belonged to category 1 “Mild”, 13 cases to category 2 “Moderate”, and 14 cases to category 3 “Severe”. Considering skin lesion distribution, 18 out of 35 were focal, and the remaining 17 were multifocal. Histology revealed a skin inflammatory reaction involving all the layers from the *epidermis* to the *subcutis*, including the underlying muscular tissue (Figure 1d–f). These were mainly characterized by: a mild to severe infiltration of lymphocytes and monocytes in the *stratum spongiosus* of the derma; a severe reaction involving the scale pockets, with evident cellular infiltration and frequent disappearance of the scales; thickening of the *stratum compactum* of the derma, heavily infiltrated by lymphocytes/macrophages; strong infiltration by lymphocytes, plasma cells, and macrophages in the *subcutis* and underlying muscular tissue. According to the criteria adopted for grading the histologic skin lesion, 12 cases belonged to category 1 “Mild”, 14 cases to category 2 “Moderate”, and 9 cases to category 3 “Severe”. Splenomegaly was the only abnormality observed on macroscopic examination of the spleen; in fact, 12 out of 35 symptomatic trout had mild to severe splenomegaly, rarely with a firm consistency and a miliary irregular appearance on the cut surface. Histologically 19 out of 35 cases belonged to category III “Hyperplastic with congestion/hemorrhage and hemosiderosis”, showing more severe modifications, including hyperplasia of ellipsoids, congestion, and hemorrhages frequently associated with hemosiderosis; 12 cases belonged to category II “Hyperplastic”, showing hyperplasia of ellipsoids and the remaining 4 cases were normal thus belonging to category I “Normal”. No macroscopical and histological changes were observed in the head kidney samples of 35 trout with RMS.

The Spearman correlation test between the morphological parameters, macroscopic category vs. microscopic category, and macroscopic category vs. multifocal distribution of the skin lesions revealed a significant correlation (R = 0.70 and R = 0.36, **** p* < 0.01, * *p* < 0.05, respectively).

### 3.2. Amplification of MLO DNA by PCR

We employed RLO1/RLO2 primers according to Lloyd et al., 2008 [19] (GenBank accession number EU555284) on the first PCR and RiFCfw–RiFCrev primers in a second step PCR according to Galeotti et al., 2017 [24]. The amplification of 188 bp after nested PCR was considered positive in this study. Skin/muscle samples were positive in 27 out of 35 symptomatic trout (77.14%). Among the 27 positive trout, sixteen fish were also positive in the tissue spleen and head kidney (Table 2).

### 3.3. Absolute Quantification of MLO Using qPCR

The standard curve generated following testing of a 10-log dilution series of the IGF and MLO plasmid had a slope of −2.975 and −3.1192. The amplification efficiency of IGF was 2.1, and MLO was 2.2. The linear correlation coefficient of IGF R2 = 0.9821 and MLO R2 = 0.9927 (Figure 2a,b). The abundance of MLO and IGF was then examined in the skin lesion, spleen, and head kidney using these quantitative PCR assays. Quantification of MLO copy number from RMS affected farm and one unaffected trout farm in different tissues has been listed in Table 3. All the fish from the affected farm in all tissues analyzed showed the amplification of MLO DNA. Seven infected skin/muscle indicated a high amount of MLO DNA compared to other tissues.

### 3.4. Immune Gene Expression

The expression of the genes analyzed in this study and related to immunity detected in skin samples are reported in Figure 3a. These samples were divided into groups based on the histological classification of skin lesions (“Mild,” “Moderate,” or “Severe”). For all the genes considered, only IL-8 and TNF-α did not show significant differences between the groups. Among the expression of cytokines (IL-1, 6, 8, 10, and TNF-α), a significant difference was found for IL-1, 6, and 10. For all cytokines, the expression in the group “Severe” was higher than in the reference group “Mild” [with high significance for IL-6 and 10 (*p* < 0.01) and good significance for IL-1 (*p* < 0.05)]. IL-1 and 10 also showed greater expression in the group “Moderate” compared with the group “Mild” (with *p* < 0.05 for IL-1 and *p* < 0.01 for IL-10). All the cell membrane receptors (TLR-5, TCR-β, MHC-I, and II) showed statistically significant differences in expression among different groups. TLR-5 showed less expression in the “Severe” group than in the reference “Mild” group (*p* < 0.01). For TCR-β, the expression was greater in the “Moderate” group than in the “Mild” group (*p* < 0.01) and in the “Severe” group (*p* < 0.01). Considering the two major histocompatibility complexes, the “Severe” group showed greater expression of MHC-I than the “Mild” (*p* < 0.01). At the same time, MHC-II was more expressed in the “Moderate” group than the “Mild” (*p* < 0.01). Both the immunoglobulins (M and T) showed the same trend, with highly higher expression in the “Moderate” group than in the “Mild” one (*p* < 0.01). The expression of genes related to immunity detected in spleen samples divided into groups “Normal”, “Hyperplastic” or “Hyperplastic with congestion/hemorrhage and hemosiderosis” is reported in Figure 3b. Among all the genes, the cytokine IL-8, the cellular receptors TCR-β and MHC-II, and the two immunoglobulins showed significant differences in expression between the groups. Considering the cytokines, only IL-8 was more expressed in the “Hyperplasia with congestion/hemorrhage and hemosiderosis” group than in the reference “Normal” group (*p* < 0.01). The cellular receptor TCR-β exhibits greater expression in both the “Hyperplasia” group (*p* < 0.01) and the “Hyperplasia with congestion/hemorrhage and hemosiderosis” group (*p* < 0.01) compared to the “Normal” group. The cell receptor MHC-II was expressed significantly differently between the two hyperplastic groups, with lower expression in “Hyperplastic with congestion/hemorrhage and hemosiderosis” (*p* < 0.01). The two immunoglobulins showed different expression between the groups considered, with IgM more expressed in the “Hyperplasia” group than in the “Hyperplasia with congestion/hemorrhage and hemosiderosis” group (*p* < 0.05), while IgT is markedly less expressed in “Hyperplasia with congestion/hemorrhage and hemosiderosis” group when compared to the “Normal” group (*p* < 0.01).

The gene expression of the immune factors in the skin, in relation to the classification of samples based on their MLO DNA content (“Low”, “Medium”, and “High”), is reported in Figure 4a. Among the cytokines, IL-10 showed a significantly higher expression in the “High” MLO DNA content group than in the “Medium” content group (*p* < 0.01), which in turn had a significantly higher expression of Il-10 than the reference group (“Low” content of MLO DNA) (*p* < 0.01). TLR-5 and MHC-II cellular receptors showed a significantly higher expression in the group with “High” content of MLO DNA compared to the group with “Low” content of MLO DNA (*p* < 0.01). Finally, both immunoglobulins showed a significantly higher expression in the “Medium” group compared to the “Low” group (*p* < 0.01). The IgT was also more expressed in the “Medium” content group than in the group with “High” content of the DNA microorganism (*p* < 0.05).

In the spleen samples, the gene expression of the immune factors considered in relation to the “Absent”, “Low”, and “Medium” categories is reported in Figure 4b. Three cell membrane receptors were expressed significantly differently between the groups; in particular, TCR-β was expressed significantly more in the “Low” and “Medium” MLO DNA content groups than in the “Absent” MLO DNA reference group (*p* < 0.01). MHC-II also had the same expression trend but with different significance levels (*p* < 0.01) when considering the “Medium” content group compared to the “Absent” one and had *p* < 0.01 for the “Low” content group compared to the “Absent” one. MHC-I showed different expressions between the “Low” and “Medium” content groups, having a higher expression in the “Low” one (*p* < 0.05). Among the immunoglobulins, IgM showed a significantly higher expression in the “Medium” and “Low” DNA content groups than the “Absent” one (*p* < 0.01).

Figure 4c illustrates the gene expression of the immune factors in the kidney in relation to the classification of samples based on their MLO DNA content. This was classified as “Very Low”, “Low” or “Medium”. Considering the cytokines, the group with the “Low” presence of MLO DNA showed a significantly lower expression than the “Very Low” content reference group for both IL-6 and TNF-α (*p* < 0.01). On the other hand, IL-1 and 10 showed significantly lower expression in the “Low” content group compared to the group with “Very Low” MLO DNA content (*p* < 0.01); for these two cytokines, there was also a significantly lower expression in the “Low group” compared to the “Medium” one (*p* < 0.05). An identical trend was also found for the cellular receptor TLR-5 expression (with the “Low” group expressing it significantly less than both the “Very Low” content group (*p* < 0.01) and the “Medium” content group (*p* < 0.05)). The MHC-II receptor revealed a significantly higher expression in the “Low” content group than in the reference group in which MLO DNA is “Very Low” (*p* < 0.01). IgM showed the same expression trend among immunoglobulins: the “Low” content group expressed it more than the “Very Low” content group (*p* < 0.05).

## 4. Discussion

Red mark syndrome represents a serious challenge for freshwater aquaculture rearing systems [1]. It can threaten land-based rearing sites in Southern Europe during cooler months [36], and the outbreaks described here exemplify a typical scenario. Various PCR methods have been developed in recent years to detect RMS-causing Rickettsia-like organisms (RLO) [19,24,37,38]. PCR and nested PCR are highly sensitive and fast and can simultaneously detect pathogens [39,40]. In this study, the primers RiFCfw–RiFCrev were used for PCR amplification of the specific product of 16S rDNA of RMS/MLO, which confirmed the disease etiology through sequencing. Our previous study also confirmed the presence of a high quantity of 16S gene copy numbers relating to trout skin lesions, suggesting that MLOs are more abundant within skin tissues of RMS in trout than in uninfected control fish [24]. Previous research indicated the presence of RLO in infected skin and spleen with variable results [3,17,24]. The first evidence of a *Rickettsia*-like organism (RLO) detected by PCR and associated with Strawberry disease (SD) (recognized the same entity of RMS in later studies) skin lesions was presented by Lloyd et al. (2008) [19]. A subsequent study on RMS detected the same DNA sequence in the skin lesion [41]. Later, a positive correlation between the quantity of RLO bacteria and the severity of the SD skin lesions was reported [20].

Moreover, an alternative novel specific absolute quantification of the MLO DNA associated with RMS was developed based on a quantitative Sybr-green real-time PCR approach [23]. Here, we tested RiFCfw–RiFCrev primers for MLO on the skin/muscle, the spleen, and the head kidney of the infected fish showing qPCR positivity for all tissues. In accordance with recent research [21,22], our study confirmed the presence of a high quantity of 16S gene copy numbers relating to trout skin lesions suggesting that MLOs are in greater abundance within skin tissues of RMS in trout but never in uninfected control fish. MLOs’ DNA localized not only within skin lesions of RMS-affected fish but also in various organs. The spleen seems involved with evident pathologic changes and the presence of MLO DNA. The fact that the spleen and head kidney showed 16S rDNA/igf1 ratio values could lead to the hypothesis that melano-macrophages in the spleen and head kidney could be primarily involved in eliminating the MLO. However, the skin remains the most important organ for understanding RMS pathogenesis.

Our study revealed a lack of correlation between the severity of skin lesions and the quantity of MLO DNA, with similar observations made in the spleen, albeit to a lesser extent. We also observed no correlation between the quantity of bacterial DNA in the skin lesions and that detected in other positive organs. Therefore, high MLO DNA in the skin did not necessarily correspond to high quantities in the other analyzed organs. These findings suggest either a variable distribution of MLO in terms of DNA quantity or a non-simultaneous dynamic of infection in different organs. Furthermore, the relatively low correlation (R = 0.36) observed between the macroscopic severity of skin lesions, and their multifocal distribution may be attributed to individual differences and variations in the timing of lesion development.

This study’s main objective was to evaluate the relationship between the expressions of several immune-relevant genes from the RMS-infected trout and the different categories of microscopic skin and splenic lesions, and the relation between MLO quantity detected in tissues and the immune gene expression. We found that in the “Severe” skin lesions, IL-1 β, IL-6, IL-10, MHC-II, and TCR were upregulated compared to the “Mild” ones. IL-1b and IL-6 are pro-inflammatory cytokines that play a fundamental role in developing inflammatory lesions. In trout, interleukin-1β is a critical pro-inflammatory cytokine that initiates inflammation during innate immune responses against bacterial and viral infections or injury [42]. Our finding agrees with another study on RMS, where IL-1β was upregulated in more severe skin lesions tested in experimentally infected fish [21]. It seems worth considering in detail the higher IL-6 expression observed in the “Severe” skin lesions. This interleukin is primarily involved in both innate and specific responses. In various teleosts, it has been demonstrated to modulate the humoral response by eliciting antibody production [43,44,45]. IL-6 can down-regulate the expression of IL-1β and TNF-α as well. Moreover, IL-6 in rainbow trout can induce the expression of hepcidin (an antimicrobial peptide) in macrophages by reducing iron availability, thus limiting the spread of infection [46]. This may suggest both a role in host defense and limit the inflammatory process: its high expression in the “Severe” cases in our study could reflect this function.

From the results presented here, fish with “Moderate” to “Severe” RMS skin lesions highly expressed IL-10, an interleukin that can suppress the inflammatory reaction counteracting the activity of IL-1β in teleosts. IL-10 shares other prototypical activities with higher vertebrates in fish, including the regulatory effect on T cells and proliferation, differentiation, and Ab secretion by IgM B lymphocytes [47]. Rainbow trout with “Moderate” skin lesions had increased IgM and IgT. Thus, IL-10 could exert a stimulatory effect by increasing local immunoglobulin production.

The high level of MHC-II gene expression was always observed in “Moderate” and “Severe” skin lesions. This is expressed in macrophages, dendritic cells, and B lymphocytes, acting as antigen-presenting cells. The increase in MHC-II expression supports the significant involvement of TCR gene expression in “Moderate” to “Severe” skin lesions reflecting an induced adaptive immune response by effector T helper lymphocytes. In light of these findings, our gene expression profile of “Moderate” skin lesions may be compatible with an adaptive immune response focused on producing antibodies. In contrast, the cell-mediated response seems predominant in “Severe” lesions. High levels of immunoglobulins and interleukins seem to control the severity of this immune response. Finally, when considering the mild skin lesion, the overall immune gene expression profile does not show an increase in any transcript, supporting an initial phase of lesion development.

When considering the relationship between the MLO DNA content in the skin and the immune gene expression, it is evident that IL-10 was upregulated in cases with “High” and “Medium” MLO DNA content, with higher values in the “High” group. These findings correlate with the severity of the histological skin lesions category where in the “Severe” and “Moderate” groups, IL-10 was upregulated, suggesting a direct role of MLO in inducing the expression of IL-10. Moreover, the skin group with “Medium” MLO DNA content had a higher expression of immunoglobulins. This was observed with the “Moderate” forms as well. We can conclude that a possible correlation between the MLO DNA content, the production of immunoglobulins, and the severity of the lesion, is present. As expected in the group of the skin with a “High” quantity of MLO DNA content, both MHC-II and TLR-5 were upregulated. MHC-II is primarily expressed on the surface of antigen-presenting cells following bacterial infections, and it can bind and present peptides derived from exogenously derived proteins to T helper cells [48]. Toll-like receptor 5 (TLR5) is a receptor for bacterial flagellin that plays a critical role in early innate immunity. In the veterinary literature, several biomolecular studies on fish response to bacterial diseases demonstrated a TLR5 overexpression [49,50]. Our study confirms an up-regulation of TLR5 in the skin with higher content of MLO DNA, suggesting that TLR5 could have an important role against the etiological agent. A recent study demonstrated the presence of a flagellar apparatus in “Candidatus Midichloria mitochondrii” [51], a member of the order Rickettsiales. Therefore, MLOs could elicit the immune response by activating TLR5 to recognize specific antigens as flagellin and induce the production of inflammatory cytokines necessary to amplify the inflammation and guide adaptive immunity.

When considering the spleen, the “Severe” lesions showed a significant increase in IL-8 and TCR and a decreased expression in IgT, whereas the “Moderate” group, characterized only by hyperplasia of ellipsoids, had a significant increase in MHC-II and TCR, reflecting a more effective adaptive immune response during infection compared to “Severe” group. The correlation between the immune gene expression and MLO DNA content in the spleen showed a significant increase in IgM and MHC-II expression and a decrease in TCR expression in fish with a “High” quantity of MLO. The same finding was observed in fish with a “Medium” quantity of MLO. As in skin, these data demonstrate a marked antibody response during infection with MLO. The beneficial effect of the antibodies could be explained by previous in vivo and in vitro studies on *Rickettsia conorii*, a bacterium with strictly intracellular growth [52,53]. These studies demonstrated the efficacy of both polyclonal antibodies against *Rickettsia conorii* and monoclonal antibodies against outer membrane proteins A (OmpA) and B (OmpB). This Fc-dependent immunity can inhibit phagosomal escape by opsonization and results in phagolysosomal killing [52,53]. Therefore, humoral immunity against rickettsiae appears crucial regarding rickettsicidial mechanisms in infected target cells. Considering the kidneys, significant data were obtained only within the group with a “Low” quantity of MLO DNA. As found in the spleen, this group presented a marked reduction in IL1, IL6, IL10, and TLR5 expression and a higher IgM and MHCII expression. Therefore, an adaptive immune response was observed with “Low” MLO DNA quantities.

This research represents a significant advancement in evaluating immune response in red mark syndrome (RMS) cases. Previous studies have explored this topic, but our integrated approach offers a novel perspective and insight into pathogenesis [21,27]. In contrast to previous investigations, we examined and sampled a larger number of fish during the same period to control for environmental variations, particularly in temperature, which significantly affects immune response development and, consequently, RMS lesions and their resolution [3,54]. Furthermore, we assessed the immune response by comparing the expression of transcripts for various genes among different groups of fish, categorized based on the severity of the histological lesion and the involvement of other organs, such as the spleen and kidney, which are considered targets for the immune response. This provided an adequate understanding of the immune profile within solid microscopic and macroscopic categories of skin lesions.

Our analysis found that, similar to McCarthy’s study [27], upregulation of IL-10 was observed in more advanced lesions. However, we took a more rigorous approach by employing statistical analyses to support the findings. In contrast to Jorgensen’s work, we investigated a spontaneous natural episode of the disease. Moreover, we correlated the immune response with the severity of the histological lesion and the involvement of the spleen and kidney. Our findings also indicated a correlation between the expression of MLO and the upregulation of IL-10 and immunoglobulins in the lesion.

## 5. Conclusions

Our study confirmed the presence of a high quantity of 16S gene copy numbers of MLO in diseased skin tissues of trout with RMS. However, the amount of MLO detected was not correlated with the severity of the skin disease. We highlighted that MLOs are confined to the skin and can invade other organs, such as the spleen and head kidney. Moreover, the spleen showed pathologic changes mainly of hyperplastic type, reflecting its direct involvement during infection. Therefore, we can state that the immune response changes according to skin and spleen disease severity and with different quantities of MLO DNA. The most severe skin lesions were characterized by a high level of inflammatory cytokines sustaining and modulating the severe inflammatory process. In the moderate form, the response was driven to produce immunoglobulins and IL-10 to control the severity of the disease. In conclusion, humoral immunity elicited during MLO infection appeared to have a fundamental role in controlling the severity of the skin disease, possibly through bactericidal antibody-mediated mechanisms.

In contrast to earlier investigations, our method enabled a thorough exploration of the host’s immune response, thus affording crucial insights into the pathogenesis of RMS. Moreover, by targeting the spleen and kidney tissues, we have, for the first time, achieved a broader comprehension of the underlying mechanisms driving this condition, which is not restricted to skin tissue and instead manifests as a systemic infection with significant involvement of splenic tissue. Our results contribute to the constantly expanding knowledge base concerning RMS and lay the groundwork for further investigations.

## Figures and Tables

**Figure 1 animals-13-01103-f001:**
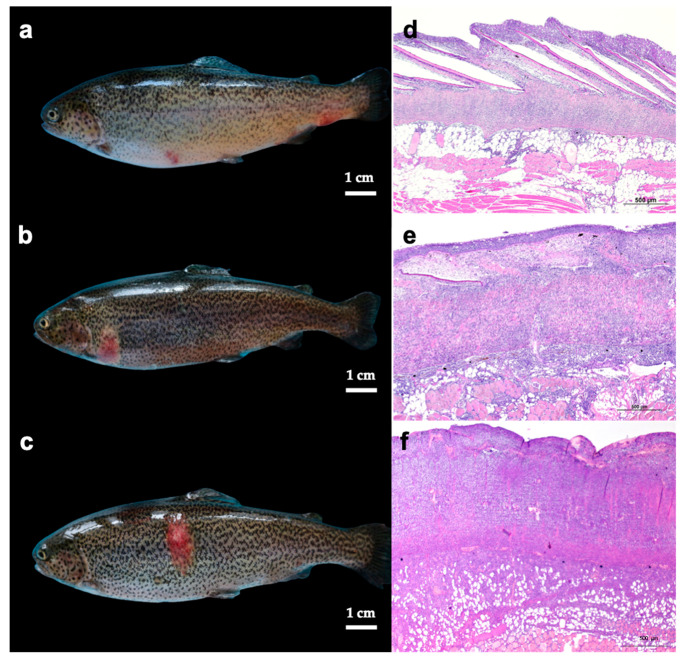
(**a**–**f**). Gross skin lesions (**a**–**c**) and their histological classification (H&E, (**d**–**f**)) included in one of three categories according to the progressive degree of severity (“Mild”, “Moderate”, “Severe”). Macroscopically, ‘’Mild’’ lesions (**a**) consist of small up to 1 cm flat macules, with discoloration, mild exudate and visible scales. ‘’Moderate’’lesions (**b**), are characterized by slightly raised 1–2 cm patches, covered by clear exudate associated with multifocal petechiae and slight scale loss. ‘’Severe’’ lesions (**c**) consist of slightly raised large patches, more than 2 cm, covered by serous/fibrin exudate with marked redness and haemorrhages, evident scale loss and area of erosion. Histologically, ''Mild'' lympho-hystiocytic inflammation (**d**) involves all the skin layers, including the hypodermis. The epidermis is intact. The scale pockets and scales are present. The cellular infiltration is particularly evident in the spongiosum layer and around scale pockets. In ‘’Moderate’’ cases (**e**), histology reveals an inflammatory response involving all the skin layers from epidermis to hypodermis, up to the underlying muscular tissue. Architecture and distinction of the layers is retained. The scale pockets and scales are not always present and scale reabsorption is often observed. The stratum compactum of the derma appears thickened with a moderate inflammatory infiltrate, recruited also towards the hypodermis and the muscular layer. Severe cellular infiltration is seen in ‘’Severe’’ cases (**f**) involving all the layers from epidermis down to the muscular tissue with epidermis partially missing. The stratum spongiosum is thinned, and no scales can be observed. The dermal stratum compactum appears thickened and severely infiltrated, deeply modifying the layer's architecture. Severe infiltration reaches the hypodermis (panniculitis) and the underlying muscular layers (myositis), spreading also between the myosepta.

**Figure 2 animals-13-01103-f002:**
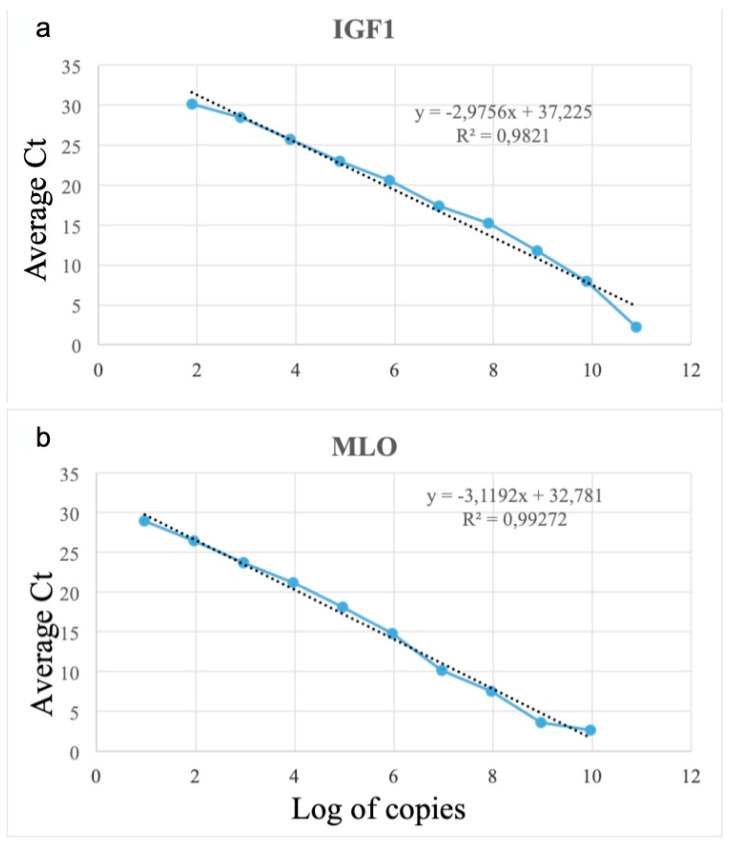
(**a**,**b**)**:** (**a**) Standard curve of SYBR, green-based RT-qPCR amplification of plasmid containing IGF segment and Midichloriaceae-like organism (**a**) Standard curve, was plotted between mean Ct values obtained from each dilution of standard plasmid IGF against calculated log copy number (slope = 2.9756, R^2^ = 0.9821). (**b**) Standard plasmid with MLO fragment against estimated log copy number (slope = 3.1192, R^2^ = 0.9927).

**Figure 3 animals-13-01103-f003:**
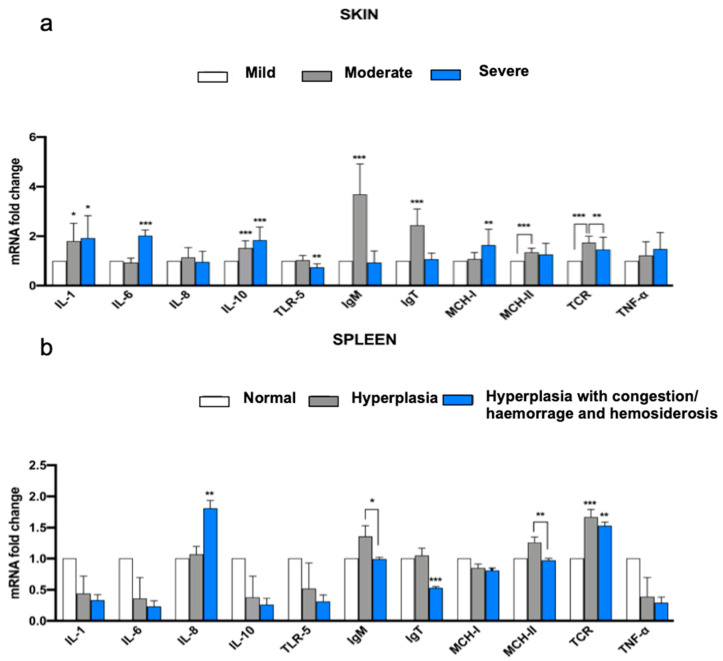
Fold change ratio in relative RNA expression of immuno-related genes analyzed in the skin (**a**) and splenic (**b**) samples. Skin and splenic samples were divided into groups based on the histological categories of the skin and splenic lesions (respectively, “Mild”, “Moderate” or “Severe” for skin and “Normal”, “Hyperplastic” and “Hyperplastic with congestion/haemorrhage and hemosiderosis” for spleen). “Mild” and “Normal” groups are the reference groups (gene expression = 1; white column) for skin and spleen samples, respectively. * = *p* < 0.1; ** = *p* < 0.05; *** = *p* < 0.01.

**Figure 4 animals-13-01103-f004:**
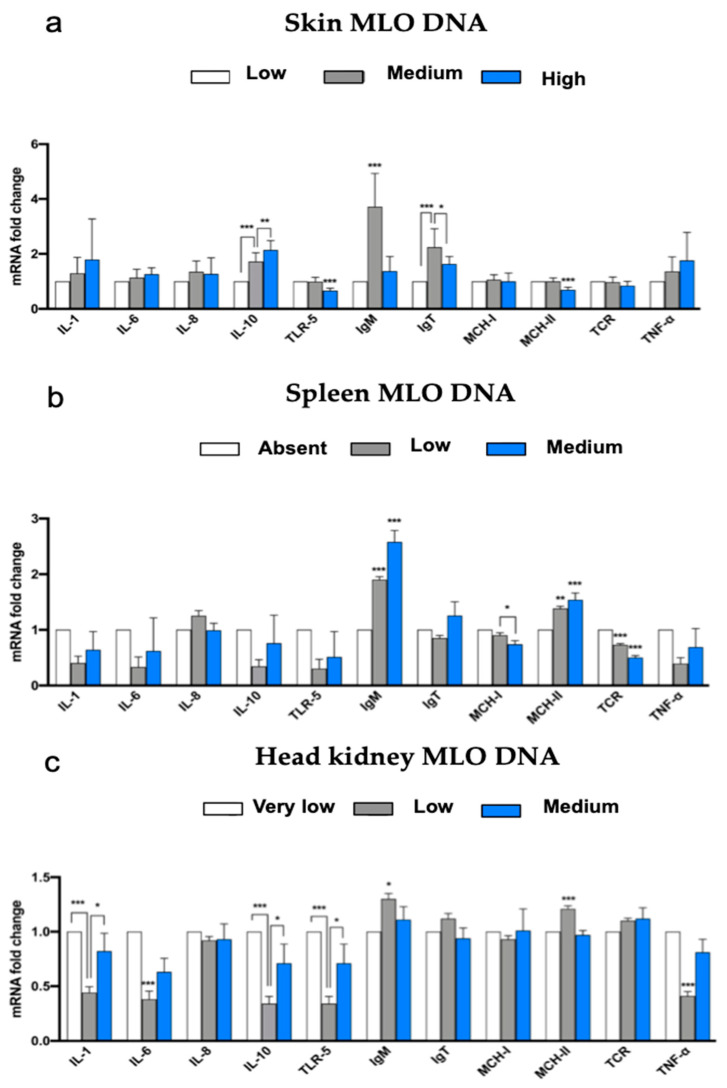
(**a**–**c**): Fold change ratio in relative RNA expression of immuno-related genes analyzed in the skin (**a**), spleen (**b**), and kidney (**c**) in relation to samples classified by their MLO DNA content. * = *p* < 0.1; ** = *p* < 0.05; *** = *p* < 0.01.

**Table 2 animals-13-01103-t002:** Results of the PCR assay performed according to Galeotti et al., 2017 [24] on the skin spleen and head kidney tissues from symptomatic and control trout using the RiFCfw–RiFCrev primers on the Amplicon obtained by RLO1–RLO2 primers with the first-step PCR.

FARM—A				FARM—B				Control			
Sample No.	Skin	Spleen	Head KIDNEY	Sample No.	Skin	Spleen	Head Kidney	Sample No.	Skin	Spleen	Head Kidney
1-12911	+	+	-	1-12931	+	+	+	C19-12929	-	-	-
3-12913	+	+	+	2-12932	+	+	+	C20-12930	-	-	-
4-12914	+	+	+	3-12933	+	-	-	C21-12949	-	-	-
5-12915	-	-	+	4-12934	+	+	+	C22-12950	-	-	-
6-12916	+	+	+	5-12935	+	+	+	C23-12951	-	-	-
7-12917	+	+	+	6-12936	-	+	+				
8-12918	+	+	-	7-12937	+	+	+				
9-12919	-	+	+	8-12938	+	+	+				
10-12920	-	-	+	9-12939	-	+	+				
11-12921	+	-	+	10-12940	+	-	-				
12-12922	+	+	+	11-12941	+	-	+				
13-12923	+	+	+	12-12942	+	-	+				
14-12924	-	-	-	13-12943	+	+	+				
15-12925	+	+	-	14-12944	+	+	+				
16-12926	+	-	-	15-12945	+	-	-				
17-12927	-	+	+	16-12946	+	+	+				
18-12928	-	+	+	17-12947	+	+	+				
				18-12948	+	+	+				

**Table 3 animals-13-01103-t003:** Quantitative PCR to detect the MLO Copy number of two RMS positive farms and one negative farm in fish tissues based on 16SrDNA/igf1 x 1000 ratio.

Farm -A				Farm -B			
Sample No.	Skin	Spleen	Head-Kidney	Sample No.	Skin	Spleen	Head-Kidney
1-12911	8.87 × 10^−3^	1.15 × 10^−4^	5.36 × 10^−4^	1-12931	1.74 × 10^−3^	1.59 × 10^−4^	8.99 × 10^−4^
3-12913	1.57 × 10^−3^	2.38 × 10^−5^	8.58 × 10^−4^	2-12932	4.27 × 10^−3^	3.94 × 10^−4^	2.93 × 10^−4^
4-12914	1.55 × 10^−2^	3.32 × 10^−3^	9.97 × 10^−4^	3-12933	5.06 × 10^−3^	6.59 × 10^−5^	2.64 × 10^−4^
5-12915	6.75 × 10^−4^	1.76 × 10^−3^	3.87 × 10^−4^	4-12934	1.42 × 10^−2^	5.28 × 10^−3^	1.63 × 10^−2^
6-12916	5.73 × 10^−3^	4.00 × 10^−5^	9.17 × 10^−4^	5-12935	1.05 × 10^−2^	-	1.05 × 10^−3^
7-12917	4.57 × 10^−3^	1.05 × 10^−3^	2.82 × 10^−4^	6-12936	3.71 × 10^−3^	4.22 × 10^−5^	2.57 × 10^−4^
8-12918	6.88 × 10^−4^	2.44 × 10^−5^	3.28 × 10^−4^	7-12937	9.11 × 10^−3^	5.59 × 10^−4^	1.42 × 10^−5^
9-12919	1.23 × 10^−3^	1.02 × 10^−3^	3.35 × 10^−4^	8-12938	8.37 × 10^−3^	5.73 × 10^−5^	2.23 × 10^−3^
10-12920	7.38 × 10^−3^	5.81 × 10^−5^	1.70 × 10^−3^	9-12939	5.14 × 10^−4^	1.33 × 10^−5^	2.07 × 10^−4^
11-12921	2.13 × 10^−4^	2.18 × 10^−5^	4.19 × 10^−5^	10-12940	3.19 × 10^−3^	6.16 × 10^−6^	1.32 × 10^−4^
12-12922	1.82 × 10^−3^	9.45 × 10^−4^	2.25 × 10^−4^	11-12941	2.85 × 10^−3^	3.83 × 10^−5^	5.38 × 10^−4^
13-12923	3.33 × 10^−3^	3.62 × 10^−5^	1.72 × 10^−4^	12-12942	8.90 × 10^−2^	3.32 × 10^−4^	1.35 × 10^−3^
14-12924	3.13 × 10^−3^	3.48 × 10^−6^	1.05 × 10^−5^	13-12943	6.40 × 10^−3^	9.38 × 10^−6^	8.99 × 10^−5^
15-12925	7.39 × 10^−3^	2.39 × 10^−3^	6.73 × 10^−4^	14-12944	1.14 × 10^−2^	7.06 × 10^−3^	-
16-12926	5.72 × 10^−4^	3.93 × 10^−5^	3.57 × 10^−4^	15-12945	3.50 × 10^−3^	3.06 × 10^−4^	3.25 × 10^−4^
17-12927	2.01 × 10^−3^	2.48 × 10^−4^	1.04 × 10^−4^	16-12946	1.51 × 10^−1^	3.72 × 10^−3^	3.96 × 10^−5^
18-12928	1.07 × 10^−4^	5.34 × 10^−6^	7.01 × 10^−4^	17-12947	2.69 × 10^−2^	1.65 × 10^−2^	1.94 × 10^−4^
				18-12948	2.14 × 10^−2^	2.30 × 10^−4^	

## Data Availability

Data are available on demand.

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
