# Peer review of "Understanding the Pathogenesis of Red Mark Syndrome in Rainbow Trout (Oncorhynchus mykiss) through an Integrated Morphological and Molecular Approach"

_animals, 2023, doi:10.3390/ani13061103_

Round 1

Reviewer 1 Report

Honestly, the pathogenesis of RMS in rainbow trout have been widely studied for many years, including cytokine expressions, histology and bacteiral isolation. The studies were shown below:

1.von Gersdorff Jørgensen, L., Schmidt, J. G., Chen, D., Kania, P. W., Buchmann, K., & Olesen, N. J. (2019). Skin immune response of rainbow trout (Oncorhynchus mykiss) experimentally exposed to the disease Red Mark Syndrome. Veterinary immunology and immunopathology211, 25-34.

2.Metselaar, M., Thompson, K. D., Paley, R., Green, D. M., Verner-Jeffreys, D., Feist, S., & Adams, A. (2020). Investigating the involvement of a Midichloria-like organism (MLO) in red mark syndrome in rainbow trout Oncorhynchus mykiss. Aquaculture528, 735485.

3.McCarthy, U., Casadei, E., Wang, T., & Secombes, C. J. (2013). Red mark syndrome in rainbow trout Oncorhynchus mykiss: Investigation of immune responses in lesions using histology, immunohistochemistry and analysis of immune gene expression. Fish & shellfish immunology34(5), 1119-1130.

Thus, I do not think that readers may be interested in the repeated research. The study should go deeper or compare the pathogenesis of different trout species.

Author Response

Dear Reviewer,

Please find attached the answers to your comments.

All the best,

Orioles Massimo

Reviewer 2 Report

The manuscript Understanding the pathogenesis of Red Mark Syndrome in 2 rainbow trout (Oncorhynchus mykiss) through an integrated 3 morphological and molecular approach is interesting and throws some insights into aquacluture enhancement by combatting red mark syndrome. The manuscript is well written with a robust methodology supported by proper results. The manuscript may be considered for acceptance after minor revisions:

1. The abstract should depict the statistically significant immune related genes, such a IL-10, and its expression compared to other genes. Proper statistical tests should be mentioned like Chi-square or regression analysis used to predict the gene that is upregulated in RMS.

2. Introduction is too short. More references need to be discussed and how your study will improve the already available data, such as audio: 10.1111/jfd.13391. Epub 2021 May 10., 86, Bull. Eur. Ass. Fish Pathol., 36(2) 2016

3. The discussion should include a subsection on how innovative techniques can lead to aquaculture enhancement and cite recent refs such as, https://doi.org/10.1016/j.aquaculture.2019.734770

Author Response

Dear Reviewer,

Please find here below the answers to your comments.

Reviewer n.2:

The manuscript Understanding the pathogenesis of Red Mark Syndrome in 2 rainbow trout

(Oncorhynchus mykiss) through an integrated 3 morphological and molecular approach is

interesting and throws some insights into aquaculture enhancement by combatting red mark

syndrome. The manuscript is well written with a robust methodology supported by proper results.

The manuscript may be considered for acceptance after minor revisions:

1. The abstract should depict the statistically significant immune related genes, such a IL-10, and

its expression compared to other genes. Proper statistical tests should be mentioned like Chisquare

or regression analysis used to predict the gene that is upregulated in RMS.

A: Thanks for your comment. The description of results regarding IL-10 and other genes and statistical analysis used were added in the abstract.

2. Introduction is too short. More references need to be discussed and how your study will

improve the already available data, such as audio: 10.1111/jfd.13391. Epub 2021 May 10., 86,

Bull. Eur. Ass. Fish Pathol., 36(2) 2016.

A. Introduction has been rewritten following your recommendations. Both the pathogenesis part and the description of RMS were expanded.

3. The discussion should include a subsection on how innovative techniques can lead to

aquaculture enhancement and cite recent refs such

as, https://doi.org/10.1016/j.aquaculture.2019.734770

A: The authors believe the study suggested is outside the scope of this research. We would be happy to discuss this further if required.

All the best,

Orioles Massimo

Reviewer 3 Report

Comments animals-2210078

Manuscript animals-2210078 is an interesting study elucidating the pathogenesis RMS in rainbow trout using combination morphological and molecular methods. Unfortunately, some major issues exist in the text which make the current form of ms is not suitable for publication.

General comments

-          Use the common name to mention the fish species, its Latin name is only included at the first appearance of species. Apply this throughout the text!

-          Ms should be rewritten for English use as many grammatical errors are found!

Simple summary

Line 17 – used

Introduction

68-77 – these two paragraphs could be combined as one!

73 – investigated

Materials and Methods

86-8 – modify the sentence!

125-7 – How was the quality of total genomic DNA evaluated? State this!

143 – It would be better to say: “PCR product size/length”.

181 – state the concentration of agarose gel applied and the dye used for RNA visualization. Same comment to the statement on line 203!

216 – were presented

Results

-          Rearrange the subsection number as section of 3.2 is missing!

275-7 – annotation letters on the figures are missing!

280-5 – Point out exactly the histological changes on the pictures!

301-2 – double check the R values!! Should be without (-)!

306 – what does the statement regarding “copy number” mean?? Clarify this!

310-4 – double check the annotation letter!! Annotation letter 3b does not appear on the picture!

316-7 – no data for negative farm found in Table 3!

334 – was more

348 – was markedly

352-5 – state the sample size, what do the Bars and asterisk mean? The same comments for Fig. 5

Discussion

428-30 – this paragraph could be developed as it only has two sentences. Normally, a paragraph consists of at least three sentences.

458-64 – what about correlation between morphological parameters, macroscopic category vs multifocal distribution of skin lesions? Low correlation (R=036) was also observed between those two parameters. This phenomenon deserves further explanations.

470 – were upregulated

485 – expressed

504 & 507 – was upregulated

513 – were upregulated

Conclusion

557 – “In conclusion” can be omitted!

Author Response

Reviewer n.3

Manuscript animals-2210078 is an interesting study elucidating the pathogenesis RMS in rainbow

trout using combination morphological and molecular methods. Unfortunately, some major issues

exist in the text which make the current form of ms is not suitable for publication.

General comments

- Use the common name to mention the fish species, its Latin name is only included at the first

appearance of species. Apply this throughout the text!

- Ms should be rewritten for English use as many grammatical errors are found!

Simple summary

Line 17 – used

Introduction

68-77 – these two paragraphs could be combined as one!

73 – investigated

A: Thanks for your comment. Each point was amended in the text. Please see new version of

the manuscript.

Materials and Methods

125-7 – How was the quality of total genomic DNA evaluated? State this!

143 – It would be better to say: “PCR product size/length”.

181 – state the concentration of agarose gel applied and the dye used for RNA visualization. Same

comment to the statement on line 203!

216 – were presented

A: Thanks for your comments. Each point was amended in the text. Please see new version of the manuscript.

Results

Rearrange the subsection number as section of 3.2 is missing!

275-7 – annotation letters on the figures are missing!

280-5 – Point out exactly the histological changes on the pictures!

These pictures have been deleted following Editor’s advice.

301-2 – double check the R values!! Should be without (-)!

306 – what does the statement regarding “copy number” mean?? Clarify this!

310-4 – double check the annotation letter!! Annotation letter 3b does not appear on the picture!

316-7 – no data for negative farm found in Table 3! A: those cases resulted negative with PCR (qualitative) and therefore were not included.

334 – was more

348 – was markedly

352-5 – state the sample size, what do the Bars and asterisk mean? The same comments for Fig. 5

A: Thanks for your comments. Each point was amended in the text. Please see new version of the manuscript.

Discussion

428-30 – this paragraph could be developed as it only has two sentences. Normally, a paragraph

consists of at least three sentences.

458-64 – what about correlation between morphological parameters, macroscopic category vs

multifocal distribution of skin lesions? Low correlation (R=036) was also observed between those

two parameters. This phenomenon deserves further explanations.

470 – were upregulated

485 – expressed

504 & 507 – was upregulated

513 – were upregulated

Conclusion

557 – “In conclusion” can be omitted!

A: Thanks for your comments. Each point was amended in the text. Please see new version of the manuscript.

Reviewer 4 Report

The manuscript includes the information about pathogenesis of Red Mark Syndrome in rainbow trout (Oncorhynchus mykiss). The authors examined fish samples form a recent outbreak and presented in detail data.

Comments:

To study the pathogenesis of diseases, models which pathogen-free animals challenged with particular bacteria are most often used. Examination of samples from recent outbreak is very valuable, but we must ensure that other pathogens have not been detected. If other studies have been done, the authors should add methods for example biochemical, molecular or serology in which the presence of other pathogens was excluded.

Line 28: The authors should add the abbreviation red mark syndrome (RMS)

Line 137: The authors should change "A nested PCR assay" to "second-step PCR".

Line 289: Please add the GenBank accession number for all own sequences of  PCR products used in analysis. The author should also provide the accession numbers of sequences from the GenBank database that were used in the analysis.

Figure 3a - Please add captions for the axes.
Figure 3b - The authors should add "3b" on the Figure.

Figure 4: The authors should increase the quality.

Author Response

Reviewer n. 4

The manuscript includes the information about pathogenesis of Red Mark Syndrome in rainbow

trout (Oncorhynchus mykiss). The authors examined fish samples form a recent outbreak and

presented in detail data.

Comments:

To study the pathogenesis of diseases, models which pathogen-free animals challenged with

particular bacteria are most often used. Examination of samples from recent outbreak is very

valuable, but we must ensure that other pathogens have not been detected. If other studies have

been done, the authors should add methods for example biochemical, molecular or serology in

which the presence of other pathogens was excluded.

A: The signs showed by the fish were typical of an RMS outbreak with no mortality, chronic skin lesion development and no impact on growth rate, feed conversion rate and appetite. The season was the classic for this disease as well. Standard bacteriological examination was performed on a very small subset of samples and did not yield significant result. There was no evident bacteria on histology as well.

Line 28: The authors should add the abbreviation red mark syndrome (RMS)

Line 137: The authors should change "A nested PCR assay" to "second-step PCR".

Line 289: Please add the GenBank accession number for all own sequences of PCR products used

in analysis. The author should also provide the accession numbers of sequences from the GenBank

database that were used in the analysis.

Figure 3a - Please add captions for the axes.

Figure 3b - The authors should add "3b" on the Figure.

Figure 4: The authors should increase the quality.

A: Thanks for your comments. Each point was amended in the text. Please see new version of the manuscript.

Round 2

Reviewer 1 Report

accept

Reviewer 3 Report

Manuscript animals-2210078 has been significantly improved by the authors according to the previous comments. I could say the current form of ms is suitable for publicaton.